# Systematic Bias in Self-Reported Social Media Use in the Age of Platform Swinging: Implications for Studying Social Media Use in Relation to Adolescent Health Behavior

**DOI:** 10.3390/ijerph19169847

**Published:** 2022-08-10

**Authors:** Sarah C. Boyle, Sebastian Baez, Bradley M. Trager, Joseph W. LaBrie

**Affiliations:** HeadsUP Labs, Department of Psychological Science, Loyola Marymount University, Los Angeles, CA 90045, USA

**Keywords:** social media use, adolescence, objective assessment, self-report assessment, validity, platform swinging

## Abstract

Public health researchers are increasingly interested in the potential relationships between social media (SM) use, well-being, and health behavior among adolescents. However, most research has assessed daily SM time via self-report survey questions, despite a lack of clarity around the accuracy of such reports given the current tendency of youth to access SM on multiple electronic devices and cycle between multiple SM platforms on a daily basis (i.e., platform swinging). The current study investigates the potential for systematic reporting biases to skew findings. Three hundred and twenty incoming college students downloaded software on their computers, tablets, and smartphones to track their active use of Facebook, Twitter, Instagram, and Snapchat over a 2-week surveillance period and then self-reported their average daily minutes on each platform immediately after. Larger proportions of students over-estimated than under-estimated their use, with the largest overestimations found on the most heavily used platforms. Females logged significantly more SM time and were less accurate in reporting than were males and, independently, the likelihood of substantial inaccuracies in reporting total SM time and time on most individual platforms increased with each additional SM platform participants reported using. Findings demonstrate that self-reported estimates of SM time among adolescents in the age of SM platform swinging are prone to substantial error and may lead to biased conclusions about relationships between variables. Alternative measurement approaches are suggested to improve the validity of future research in this area.

## 1. Introduction

Social media (SM) has become an increasingly important aspect of adolescence, providing opportunities for bonding and bridging social capital, peer interaction, identity expression, and connection to others outside of their immediate environments [1,2,3]. Yet, research has also highlighted potential dark sides of SM, with studies linking greater time on SM to a myriad of negative outcomes, including depression [4,5]), body image concerns [6,7]), sleep disturbance [8,9], alcohol use [10,11], misperceived health behavior norms [12,13], and cyberbullying victimization [14]. However, most studies examining these relationships have relied on survey items to retrospectively assess daily or weekly on a specific SM platform or across all SM used, acknowledging that the accuracy with which individuals are able to retrospectively self-report these behaviors is unclear. While commonly identified as a study limitation, other recent research has argued that reporting errors in the domain of SM time are likely to be random rather than systematic [15], and therefore should not be of great concern to the researchers asking these questions. Given the increasing interest in the psychosocial and behavioral correlates of SM use during adolescence, the current study furthers this debate using improved methods. Consistent with adolescents’ present tendency to regularly use multiple SM platforms on multiple electronic devices [16,17], this research employs cross-device objective time-tracking software to evaluate and compare the validity of young adults’ self-reported daily time on four widely used SM platforms. Also examined are three novel factors that may be systematically associated with an increased likelihood of inaccurate self-reports in the current SM landscape: the particular SM platform in question, the sex of the participant, and the total number of SM platforms used by the participant.

### 1.1. Previous Research and Limitations

Among the first to consider the potential for inaccuracy in retrospective reporting of SM time was Junco [18]. A software program was installed on the personal computers of participating college students to monitor the minutes per day they spent on Facebook over a 30-day period. Results revealed that students substantially overestimated their Facebook use, self-reporting an average of 145 min per day of active use while objectively logging only an average of 26 min. This study was limited in that it objectively tracked Facebook time on students’ computers but not on other devices (i.e., smartphones, tablets), and included a small sample (*N* = 49) of mostly female (73%) students preventing examination of sex differences. Moreover, fewer than half of the students who completed the baseline survey (which comprised only 17% of students to whom it was sent) agreed to install the computer tracking application, further limiting generalizability. In recent years, several studies have revisited the issue of self-report accuracy in the context of SM use, generally replicating two findings with larger samples: reporting inaccuracies are common, and they tend toward the direction of over-reporting [19,20]. However, these studies share substantial limitations with the initial investigation [18] including the assessment of SM time on one specific electronic device (iPhones) [20], a single platform [19], or lumping time on multiple SM platforms together to examine total time [15,20]. Due to these issues, findings and implications are limited in their usefulness to researchers designing new studies. For instance, in single device studies, some portion of participants’ reporting errors may be attributed to difficulty cognitively separating SM time on a specific device (i.e., iPhone) from total SM time across electronic devices (e.g., iPhone, tablet, and computer). Thus, participants may report total time across devices while only time on a specific device is tracked, artificially leading researchers to observe over-estimation. Further, findings from studies investigating self-report accuracy for total time across a broad category of SM have unclear implications for researchers who tend to be interested in the use of specific SM platforms [6,21] in order to inform the development of platform-specific health intervention or prevention efforts [22,23]. Previous studies require the assumption that self-report accuracy is consistent across SM platforms used more and less frequently used by adolescents, despite some evidence that heavier SM use affords greater opportunity for reporting an error [20]. Additionally, few studies have investigated platform or participant-level factors that may increase the likelihood or degree of inaccuracy or over-reporting. Greater insight into these factors may help public health researchers investigating adolescents’ social media time optimize their methods.

### 1.2. Explanations for Reporting Inaccuracy and Potential Systematic Predictors

Social desirability [24,25], careless responding [26], the construction of self-report assessments [27,28], and social identity-related self-perception [29,30,31] are common explanations for measurement error in survey research with adolescents. Although youth are likely to perceive adult researchers to find their spending less time on SM more desirable than more time, adolescents and emerging adults have been found more likely to over-report this behavior, suggesting that social desirability concerns may not explain SM reporting inaccuracies well. However, recent findings suggest that careless responding may play at least some role in the inaccuracy of SM use estimations among adolescents. Sewall [20] highlighted associations between depressed mood, heavier iPhone-specific SM use, and larger discrepancies between objective assessment and self-reported iPhone-specific time on SM. Thus, the depressed mood appears to be one factor that may lead to inaccuracy potentially through a lack of motivation. Building on this work, this research examines how identity processes, self-perception, and use of specific SM platforms may also be implicated in the accuracy of self-reported use time and the direction of discrepancies.

#### 1.2.1. SM Platform

Recent surveys of American adolescents have revealed patterns of frequent and heavy use of Snapchat and Instagram coupled with dwindling Facebook and Twitter engagement [32,33,34,35].. Meanwhile, research in the domain of behavior measurement suggests that self-reports of behavioral data are often inaccurate especially if the behaviors in question comprise habitual and indistinct events such as frequency of anger or smartphone use [36,37]. Given that recent data suggests that young adults may engage in the habitual use of some SM platforms (Instagram, Snapchat) more than others (Facebook, Twitter), it follows that there may also be platform-based differences in the accuracy with which students self-report their use. Likewise, Junco [18] proposed that students’ estimates of their Facebook use may have been inflated by stereotypes and media depictions of young adults as heavy Facebook users. However, since 2013, stereotypes around SM use have shifted considerably with Facebook now recognized as tremendously popular with older adults, and consequentially, less frequented by young people [38]. In contrast, today’s news stories, song lyrics, and television shows more frequently reflect the popularity of Instagram, Snapchat, and other image-based platforms among youth [39,40] potentially making young adults more likely to overestimate their own use of these popular SM platforms. This prediction is consistent with the finding that behaviors recognized to be normative in the peer group are commonly over-reported in surveys due to identity processes and self-perception [29,30,31].

#### 1.2.2. Reporter Sex

Participant sex and gender may similarly impact the reliability of reporting on SM-related behaviors through identity-related processes. American females reliably self-report more time on SM than do males [41,42,43]. However, it remains unknown whether these sex differences in self-reported SM use coincide with objective reality or reflect artifacts of gendered self-perception. For instance, research suggests that females devote more of their online time to maintaining and cultivating interpersonal relationships whereas males tend to be more focused on information-seeking and non-social tasks [44]. Further, females tend to derive a greater sense of self through the strengths of their social ties and relationships than do males [45]. Thus, it stands to reason that heavy SM use may be perceived as more normative among females than males, potentially making females more likely to overestimate their time on SM.

#### 1.2.3. SM Platform Swinging

The majority of young adults today engage in platform swinging, a term coined by Tandoc and colleagues [3] to describe the regular use of multiple SM platforms and the routine rotation or cycling of engagement between platforms. As a large proportion of the SM research conducted to date has focused on one platform (i.e., Facebook or Instagram), there has been very little consideration of platform swinging’s potential cognitive effects. For instance, it is unknown whether platform swinging reflects habitual behavior or more deliberate and conscious processes. Also unknown is the extent to which platform swingers are able to cognitively distinguish between their time on different platforms. Thus, with regard to SM research methods, a critical question lies in whether platform swinging impedes young adults’ ability to accurately report their time across the multiple SM platforms they use. That is, just as multitasking on mobile devices has been found to increase recall biases for different app usage categories [46,47] cycling engagement between a greater number of SM platforms may result in decreased accuracy when self-reporting time on individual platforms.

### 1.3. The Current Study

Building on Junco’s initial investigation [18], this study recruited a large sample of incoming college students and utilized a novel software application to track students’ active minutes per day using Facebook, Instagram, Snapchat, and Twitter across their electronic devices (personal computers, tablets, smartphones). Informed by Junco’s findings, we predicted that students would exhibit sizable inaccuracies in their self-reported use across platforms (**H1**) with students more likely to overestimate than underestimate their daily use (**H2**). Guided by research on the impact of identity and self-perception in the survey measurement context, poorer accuracy and greater over-estimations of use were expected for Instagram and Snapchat, relative to older platforms Facebook and Twitter, due to heavier actual use of these newer platforms by participants and media representations of young adults as heavy users (**H3**). Extending the same logic, we also predicted that female students would be more prone to inaccuracies and more likely to overestimate their SM use than male students (**H4**). Finally, we hypothesized that self-reporting use of a greater number of SM platforms during the surveillance period would increase the likelihood of substantially over-estimating time across SM platforms and on any given platform (**H5**).

## 2. Methods and Materials

### 2.1. Participants

Participants were 320 matriculating college students recruited as part of a larger study investigating SM influences among incoming college students at a mid-sized university in the western United States. Recruitment took place over a two-week period in July of 2018, prior to matriculation. To be eligible for the larger study, students had to be registered as an incoming first-year student, 18 years of age or older, in possession of an Apple or Android smartphone, active on at least one SM platform, and planning to live on campus during their first year. Interested students meeting eligibility requirements were invited to complete baseline assessments until the 320 spots in the study were filled, which occurred in approximately one week. The study’s sample was representative of the freshman class at the host university as the mean age of the sample was 18.60 years (*SD* = 0.26), 63% were female, 48% were Caucasian, 15% were Asian, 10% were African American, 17% were Hispanic, and 9% were multi-racial or other.

### 2.2. Recruitment and Study Procedures

Prior to the start of the fall semester, the Registrar’s office emailed incoming first-year students a link to the larger study’s informational website, which detailed all study procedures and included a link to a screening survey that allowed interested students to assess their eligibility. Eligible students were invited to participate and, after providing informed consent, completed a baseline survey. At the end of the baseline survey, participants were prompted to install a custom research version of the software application, RescueTime, onto their smartphones, tablets, and personal computers to track their Facebook, Instagram, Snapchat, and Twitter use over a two-week surveillance period. Blind to the RescueTime data being collected from their devices during this time, participants were texted and emailed a link to complete a short survey at the end of the two-week period. The first page of the survey prompted students to estimate their average daily minutes spent on each of the four SM platforms (Facebook, Instagram, Twitter, and Snapchat) over the previous 2 weeks. Upon submission, participants with Android phones were directed to a personalized dashboard at which they could view their RescueTime SM time tracking data. However, as the RescueTime software application was not fully compatible with iPhones, participants with iPhones were first directed to a survey page that provided instructions for navigating to the application battery usage screen of their iPhone, using battery screen settings to display active SM application use data in minutes per day over the previous 2 weeks, taking a screenshot of the data displayed, and uploading this screenshot into the survey. Following submission, they advanced to the RescueTime dashboard. The university’s Institutional Review Board approved all study procedures and measures.

### 2.3. Measures

#### 2.3.1. Demographics

Participants reported basic demographic information including sex, age, race, and ethnicity.

#### 2.3.2. Objective Daily Time on SM

A custom research version of RescueTime, a commercially available cross-device time management software program, was used in this study to track participants’ total minutes of active Facebook, Instagram, Snapchat, and Twitter use on their personal computers, tablets, and smartphones. Participants were required to install the custom application on at least two electronic devices (i.e., personal computers, tablets, smartphones) to track their SM use over a surveillance period spanning the first 2 weeks of August, prior to matriculation. During the surveillance period, participants could not view any of their RescueTime use data. Meanwhile, a data dashboard allowed the research team to check device installs for each participant and export CSV files detailing each participant’s total minutes spent actively using Facebook, Instagram, Snapchat, and Twitter applications and related websites across devices. Following the surveillance period, this data was exported and the total minutes of active use on each SM for each participant were divided by 14 to calculate an objective measure of average daily minutes spent on each SM platform during the 2-week period. In addition, average daily minutes on each SM were summed to derive an objective measure of total minutes per day on SM.

#### 2.3.3. Supplemental Battery Screen Data (iPhone Owners Only)

As the RescueTime software application was not fully compatible with iPhones, iPhone owners were required to follow additional steps to ensure that the time they spent actively using the four SM platforms from their phones during the surveillance period could be combined with the data from their other electronic devices logged by RescueTime. After self-reporting their SM use at the end of the surveillance period, iPhone users were directed to a second survey page which provided detailed step-by-step instructions for navigating to the application battery usage screen of their iPhone, manipulating the battery screen settings to display active SM application use data in minutes per day over the previous 2 weeks, taking a screenshot of the data displayed, and uploading this screenshot into the survey. Battery screen data documenting each iPhone user’s total minutes of active Facebook, Instagram, Snapchat, and Twitter IOS app usage over the 14-day period was pulled from these screenshots and added to each participant’s RescueTime derived total minutes of active use across their other devices (computers, tablets), prior to computing average daily use for each platform.

#### 2.3.4. Self-Reported Platform Use and Daily SM Time

Immediately following the surveillance period, participants were sent a brief survey that prompted them to indicate the SM platforms they actively used during the previous 2 weeks from a list that included Facebook, Twitter, Snapchat, and Instagram. Then, for each platform they reported using, participants were prompted to estimate their average minutes per day of active use over the previous 2 weeks. Platform-specific daily minute estimates were used individually in analyses and were also summed to create a composite measure of self-reported daily total time on SM.

### 2.4. Analytic Plan

First, patterns of SM platform use according to self-report and objective measures are examined via descriptive statistics and *t*-tests. Then, the prediction that students would exhibit significant inaccuracies in estimating their daily SM use (H1) is dually examined via paired sampled *t*-tests focused on discrepancies between measures of SM time and bivariate correlations between measures. Next, tests of directional hypotheses H2–H5, which focus on the likelihood of problematic overestimation versus underestimation versus accurate reporting of daily SM time, are evaluated by more meaningfully classifying participants based on the size and direction of their reporting discrepancies. Cut-points used to classify participants were informed by the distribution of reporting discrepancies in the sample as well as consideration of the degree of error that may be meaningful in the applied health research context. For each platform and across platforms, participants who self-reported use within ±9 min of their objective assessment are classified as “Accurate”, those who self-reported time 10 or more minutes less than their objective assessment are classified as “Underestimators”, and those who self-reported time exceeding their objectively logged time by 10 min or more are classified as “Overestimators”. The predictions that participants would be more likely to overestimate than underestimate their use overall (H2) and be more likely to overestimate their daily time on newer platforms relative to older platforms (H3) are evaluated by comparing proportions of participants classified as Underestimators, Overestimators, or Accurate Reporters via Chi-Square Goodness of Fit tests. Then to evaluate H4 and H5, multinomial regression models predicting participant accuracy classification (i.e., underestimator, overestimator, or accurate reporter) are examined for each SM platform and across all SM as functions of participant sex (H4) and the total number of SM platforms participants reported using (H5).

## 3. Results

### 3.1. Preliminary Analysis

Figure 1 compares the “users” of each SM platform in terms of the percentages of participants who self-reported active use versus objectively logged minutes of active use. Striking here is the observation that greater numbers of participants self-reported active use of Facebook (*n* = 289) and Twitter (*n* = 199) than objectively logged any time on these platforms (Facebook *n* = 206; Twitter *n* = 150) during the surveillance period. Meanwhile, for Instagram and Snapchat, which were more widely used in the sample, similar numbers of participants self-reported (Instagram *n* = 304; Snapchat *n* = 297) and objectively logged use (Instagram *n* = 301; Snapchat *n* = 296).

As proportionally depicted in Figure 2, the number of SM platforms used by participants significantly differed by assessment type, *t*(318) = 12.14, *p* < 0.001, with the number of SM platforms used according to self-report (*M* = 3.41, *SD* = 0.77) exceeding those used according to objective data (*M* = 2.97, *SD* = 0.87). By sex, male (*M* = 3.38, *SD* = 0.84) and female (*M* = 3.43, *SD* = 0.72) participants did not significantly differ in the number of SM platforms they self-reported using, *t*(318) = −0.61, *p* = 0.54. However, objectively, males (*M* = 2.84, *SD* = 0.95) logged time on significantly fewer platforms than did females (*M* = 3.06, *SD* = 0.80).

Table 1 presents raw means and standard deviations for objectively assessed and self-reported minutes per day on each SM. Within sets of self-reports and objective assessments, paired samples *t*-tests examined platform-based differences in minutes per day of active use with significant differences flagged in the overall column. Consistent with recently published SM trends among adolescents and young adults based on self-reported data (Auxier & Anderson, 2021), both sexes self-reported and objectively logged greater daily time on Snapchat and Instagram than they did on older SM platforms, Facebook and Twitter. Within measurement sets, independent samples *t*-tests also examined sex differences in minutes per day of use by SM platform and overall. Consistent with previous self-report findings, relative to males (*M* = 87.84, *SD* = 75.85), females objectively logged (*M* = 110.60, *SD* = 99.32) significantly greater daily time on SM, *F*(1, 318) = 4.69, *p* = 0.03, with females’ greater time on Instagram and Snapchat primarily driving these differences. A parallel but slightly more extreme pattern of sex differences was observed among self-report estimates.

### 3.2. Accuracy of Self-Reported Daily SM Time (H1)

Tests of the mean differences between self-report and objective measures are presented at the bottom of Table 1. Indicating support for H1, overall participants exhibited significant discrepancies in their self-report estimates of SM time overall and on each platform. Although measurement discrepancies were bidirectional, the negative values of discrepancies indicate that the average participant overestimated, rather than underestimated their use of each platform. Correlations provided in Table 2 between self-report and objective measures of daily SM time overall and by sex offer additional support for H1. Tests of differences between male and female-specific correlations by platform and overall are presented in the right-most column of Table 2. The relationship between self-reported and objectively assessed total daily SM time was significantly weaker among females than males (*Z* = 3.79, *p* < 0.001), and this was primarily driven by females’ weaker correlations between measures of daily Instagram (*p* = 0.01), Snapchat (*p* = 0.003) time. Thus, females were less accurate than males in reporting their total daily SM time, and this was primarily driven by females’ poorer accuracy in reporting their Instagram and Snapchat time.

### 3.3. Major versus Minor Inaccuracies in Self-Report Overall (H2) and by SM Platform (H3)

In contrast to raw discrepancies and correlations between measures, our classification of participants as accurate reporters, under-estimators, and over-estimators more meaningfully identified the proportions of participants who substantially and problematically misreported their SM time in either direction relative to those who were inconsequentially off only a handful of minutes (See Figure 3). In support of H2, a significantly larger percentage of participants were substantially over-estimated (59%) than substantially under-estimated (23%) or accurately reported (18%) their total SM time, *X*^2^(2)= 95.14, *p* < 0.001. Providing support for H3, relative to the numbers of participants who substantially over-estimated their time on older, lesser-used platforms, Facebook (*n* = 80, 25.1%) and Twitter (*n* = 85, 26.6%), nearly twice as many over-estimated their time on newer, more heavily used platforms, Instagram (*n* = 145, 45.5%) and Snapchat (*n* = 157, 49.2%), all *p*s < 0.001.

### 3.4. Is Sex (H4) or # of SM Used (H5) Related to Likelihood of Over/Under-Estimation?

Table 3 presents results from multinomial regression models examining the relative impacts of female sex and the number of SM platforms participants reported using on the likelihood of substantially under-estimating or over-estimating (relative to accurately reporting) total SM time and time on each individual platform.

Suggesting a lack of support for H4, female sex was not significantly associated with over-estimating (relative to accurately reporting) total daily SM time when the number of SM platforms used was held constant. However, being female was associated with increased odds of over-estimating daily time on Instagram and both over-estimating and under-estimating time on Snapchat. Meanwhile, female sex was not related to the likelihood of over-estimating or underestimating time on less used platforms Twitter or Facebook. In support of H5, holding constant sex, reporting use of a greater number of SM platforms was related to significantly greater odds of substantially over-estimating (relative to accurately reporting) total daily time on SM. Results varied somewhat by individual platform. Consistent with H5, using a greater number of platforms was associated with an increased odds of substantially over-estimating daily time on Instagram. Meanwhile, using a greater number of SM platforms was significantly associated with increased odds of both substantially over-estimating and under-estimating daily time on Snapchat and Twitter. In sum, as the number of SM platforms participants reported using on a daily basis increased, so did the likelihood of committing substantial inaccuracies in self-reporting total SM time as well as time on 3 of 4 the individual SM platforms of focus in this study.

## 4. Discussion

This study compared objective cross-device SM time-tracking data to self-reported daily SM time in order to investigate whether systematic reporting biases may obscure public health researchers’ understanding of psychosocial and behavioral correlates of SM use. Concerningly, many participants in this study were unable to accurately report the number of SM platforms they used during the 2-week surveillance period, reporting use of Facebook and Twitter during this period despite objectively spending no time at all on these platforms across their electronic devices. Further, examining the accuracy of self-reported daily time spent on four popular SM platforms individually across electronic devices yielded results largely consistent with those from previous studies focused on the use of a specific SM platform on a specific device [18]: inaccuracies were abundant and tended to be in the direction of over-estimation. However, in contrast to recent findings suggesting that inaccuracies in self-report are random and should not be of concern in survey research [15], results from this study revealed the likelihood of meaningful overestimation to be a function of both specific SM platform in question and participant sex. As expected, participants were more likely to overestimate time spent on the platforms they used more heavily (e.g., Instagram and Snapchat) relative to those used less (e.g., Facebook and Twitter). Thus, platform-based differences in the accuracy of self-report may simply be a function of daily time spent on a platform (e.g., spending greater time on Instagram and Snapchat may make it more difficult to accurately report daily time on these platforms) rather than cognitive factors or platform characteristics. However, these findings are also consistent with the implicit theory explanation introduced by Junco [18] nearly a decade ago as well as sources of survey measurement error rooted in social identity and peer norms [29,30,31]. That is, because pop culture depictions of young people as particularly heavy users of Instagram and Snapchat are plentiful, adolescents’ estimates of their daily time on these platforms may also be inflated by perceptions of normative use among in-group peers. Similarly, results revealed that females spent more time than males on SM platforms considered normative amongst youth (Instagram and Snapchat) and are also more likely to overestimate their daily time on these SM. These findings are not surprising given that these SM platforms may help satisfy psychological needs that are typically stronger in females than males: cultivating interpersonal relationships [44] and deriving a greater sense of self through social ties [45]. Thus, females’ greater tendency toward over-estimation may be explained both by their greater time spent on particular SM and perceived gender-specific norms regarding the use of those platforms.

Given the ever-expanding SM landscape and adolescents’ tendency toward platform swinging [3], this study’s most important finding may be the link between the number of SM platforms used during the surveillance period and the accuracy of self-reported time on individual platforms. In sum, as the number of SM platforms participants reported using increased, so did the likelihood of committing substantial inaccuracies in self-reporting total SM time as well as time on 3 of the 4 individual SM platforms assessed. Thus, it appears that platform swinging does indeed impede young adults’ ability to accurately estimate the time they spend on individual SM platforms and that cognitive bleed may occur between these online activities. These findings are consistent with those from previous studies linking greater cycling between mobile applications to greater recall bias [46,47] and taken together suggest that the phenomenon of misreporting time spent on SM may be a broader issue associated with multi-tasking on electronic devices.

### 4.1. Implications for Researchers Studying Social Media Use among Adolescents

This study was designed to closely map onto the self-report validity concerns of researchers increasingly interested in the degree to which cross-device time on one or more popular SM platforms is associated with the well-being or health behavior among adolescents. Findings suggest that self-report validity concerns in this literature are very much warranted as inaccuracies in self-reported SM time were found to be systematically related to the specific SM platform in question, participant sex, and the number of SM platforms used regularly by participants. These findings are concerning as such systematic biases may obscure the true relationships between variables. For instance, the larger project from which the current investigation was derived examined prospective relationships between daily time on the four SM platforms during the transition into college in relation to subsequent perceptions of peer drinking norms and drinking behavior later in the school year. When objective data were used in analyses, as planned, the theorized model was fully supported [48]. However, when the model was re-run using self-report survey data from the same participants several key linkages were no longer supported (results available upon request) presumably due to the noise created by the reporting biases identified in the present study. Underscoring the importance of objective assessment in the SM use domain, reliance on only self-report measures would have led to the incorrect conclusion that daily SM time is not a predictor of drinking over the first year of college. Fortunately, compared to Junco’s initial study [18], where less than half of the participants who completed the baseline assessment installed the time-tracking software on their computers, the current sample was easily recruited despite the multi-device objective time-tracking requirements and no participants withdrew from the study due to this or other SM privacy invasive aspects [21]. This suggests that SM privacy concerns among youth may have faded considerably since 2013 and should encourage researchers to find cost-effective ways to objectively track SM platform time across adolescents’ electronic devices.

Although supporting a shift toward objective SM time tracking methods, findings also carry implications for researchers limited to the use of self-report measures. That is, a greater understanding of the demographic, personality, and social identity characteristics systematically related to over-reporting SM time may help researchers for whom an objective assessment is not possible to reduce or account for such reporting biases. For example, findings suggest that the accuracy of self-report is higher on SM platforms used less heavily (e.g., Facebook, Twitter). Thus, researchers limited to self-report may prefer to investigate time on these platforms. Alternatively, rather than ask multiple choice or free response questions about daily or weekly time on SM platforms prone to systematic reporting bias, it may be optimal for researchers limited by survey-based assessments to adopt validated, psychometrically sound scales to assess SM addiction or problematic SM use [49,50] in efforts to study correlates of excessive use. However, for survey researchers who need to assess daily time using a specific popular SM platform rather than SM addiction or problematic use, findings from this study suggest that it may be useful to query participants about their time on various SM rather than only ask questions about the particular SM platform of interest. Although more research is needed, it may be possible to reduce reporting-related noise in relationships of interest in the analysis stage through advanced statistical techniques to the extent that associated participant-level data (i.e., sex, number of SM platforms used, and time on other SM platforms) is available.

### 4.2. Limitations and Future Directions

This study is limited in that data came from a sample of incoming first-year students from a single American university. However, the sample size exceeded those utilized in previous studies [15,18], and self-reported daily SM platform use patterns mapped onto national survey data from American 18–24-year-olds [32,49], suggesting that findings may generalize to older American adolescents beyond the current sample. However, it is unknown whether the similar patterns of daily use and systematic reporting biases generalize to younger adolescents or adolescents outside of the United States. An additional limitation was related to the RescueTime software application used in this study not being compatible with iPhones. This resulted in iPhone users being required to complete additional steps to ensure that the time they spent actively using the four SM platforms from their phones during the surveillance period could be combined with the data from their other electronic devices. To avoid such an extra burden among a subset of participants, it will be important for future research to develop and utilize cross-device software programs able to track participants’ SM use more seamlessly across all versions of IOS, Android, and Windows-based operating systems.

Another potential limitation is that participants’ awareness of the software installed on their electronic devices may have impacted their patterns of SM use to some unknown degree. However, data were collected within a larger multi-component project which likely worked against the potential for any such software impact. That is, participants were invited to take part in a 9-month study investigating the potential psychological impacts of what they see on SM during the transition into college. Informed consent information framed the collection of this study’s data in the context of the larger project’s broader research questions (e.g., “does seeing a lot of posts about partying or studying on SM influence your perceptions of how much college students really party or study?”). Although participants installed the tracking software and reported on their SM use in one of five surveys, in the service of the larger project they also provided their SM account logins so a bot could periodically sample the posts in their social media feeds, and were repeatedly surveyed about their normative perceptions, and completed multiple implicit association tests. Participants were also encouraged to use social media as usual during the study period and the research team’s secondary interest in their reporting accuracy was not disclosed. Although participants’ patterns of SM use and the discrepancies observed between objectively tracked and self-reported time seem unlikely to be an artifact of the time-tracking software or broader study participation, additional attention to this issue is warranted.

A final limitation is that the study did not objectively assess participants’ time spent on every SM platform currently available (e.g., TikTok, YouTube, etc.), and instead narrowed its focus to the four platforms most widely used by American adolescents at the time of data collection and commonly studied in relation to well-being and health behavior outcomes. Although more research is needed, this study’s findings suggest that expanding focus to a larger number of popular SM platforms may lead to even greater reporting inaccuracies among adolescents swinging a greater number of platforms, with these inaccuracies potentially further obscuring true relationships between time on SM, well-being, and health behavior. Thus, examining self-report inaccuracies on additional SM platforms and evaluating potential correlates such as culture, social identity, perceived in-group norms for platform use, and other SM variables (e.g., motivations for use, passive lurking versus active sharing or interaction) represent important directions for future research that may provide researchers investigating SM use among adolescents additional insight into survey question optimization and illuminate novel methods to reduce bias in self-report data.

## 5. Conclusions

Despite public health researchers’ tendency to assess adolescent social media behavior via self-report survey questions, findings from this study reveal that many adolescent participants are unable to accurately report the SM platforms they used over the previous 2 weeks, and that, on average, adolescents also substantially over-estimate the average daily time they spend on individual SM platforms over this period of time. Further, the accuracy of adolescents’ responses to questions about average daily use appears to be systematically associated with the specific social media platforms in question, as well as the participant’s sex and the total number of social media platforms they use regularly, thereby biasing the results of survey-based studies investigating health-related correlates of SM use. As understanding adolescent patterns of SM use and psychosocial and behavioral correlates represent critical areas of inquiry that may inform the development of new public health intervention and prevention programs, the validity of social media assessments must improve.

## Figures and Tables

**Figure 1 ijerph-19-09847-f001:**
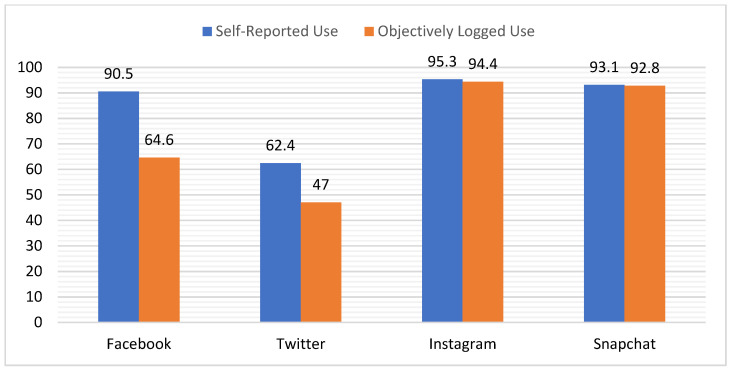
Percentages of the sample self-reporting versus logging active use of each SM.

**Figure 2 ijerph-19-09847-f002:**
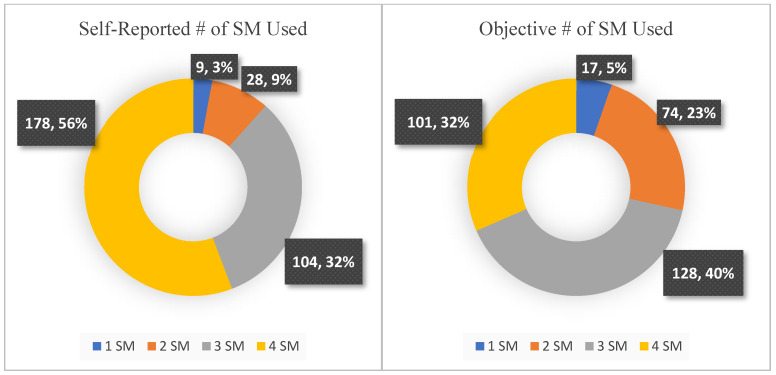
Number (#) of SM platforms “used” assessed via self-report and objectively logged time.

**Figure 3 ijerph-19-09847-f003:**
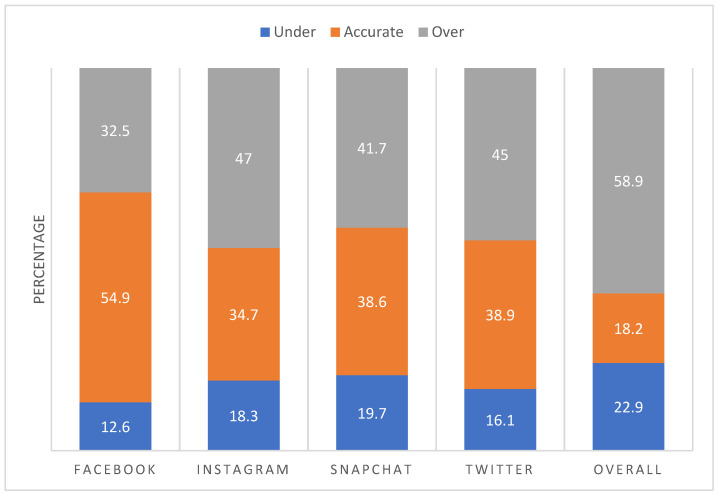
Percentages of students under-reporting, accurately reporting, and over-reporting their social media use by platform and across all platforms among all participants.

**Table 1 ijerph-19-09847-t001:** Objectively assessed and self-reported daily minutes of active SM use and discrepancies between measures overall and by sex.

	Overall (*N* = 319)	Males (*N* = 121)	Females (*N* = 198)
	*M* (*SD*)	*M* (*SD*)	*M* (*SD*)
** Objective Measure **			
Facebook ^A^	11.49 (18.26) ***^C, D^	10.34 (20.51)	12.19 (16.75)
Twitter ^B^	14.32 (33.65) ***^C, D^	14.25 (29.01)	14.36 (36.20)
Instagram ^C^	34.42 (37.49) ***^A, B^	27.60 (28.12)	38.58 (41.71)
Snapchat ^D^	41.75 (43.09) ***^A, B^	35.65 (36.43)	45.46 (46.37)
Total SM	101.99 (91.70)	87.84 (75.85)	110.60 (99.32)
** Self-Report Measure **			
Facebook ^A^	18.28 (30.12) ***^C, D^	17.00 (29.58)	19.07 (30.49)
Twitter ^B^	20.69 (32.75) ***^C, D^	18.93 (29.35)	21.76 (34.69)
Instagram ^C^	49.96 (52.12) ***^A, B^	31.56 (27.82)	61.21 (59.82)
Snapchat ^D^	57.51 (68.08) ***^A, B^	42.31 (43.18)	66.80 (78.21)
Total SM	146.38 (129.54)	109.80 (87.01)	168.74 (145.44)
** Mean^diff^ **			
Facebook ^A^	−6.83 (28.62) ***	−6.65 (27.76) **	−6.93 (29.21) ***
Twitter ^B^	−6.34 (35.68) **	−4.70 (28.89)	−7.34 (39.29) **
Instagram ^C^	−15.53 (55.28) ***	−3.95 (29.72)	−22.60 (65.28) ***
Snapchat ^D^	−15.98 (65.10) ***	−6.66 (37.25)	−21.67 (76.87) ***
Total SM	−44.60 (137.79) ***	−21.95 (78.50) **	−58.45 (162.44) ***

*Notes*. Objective and Self-Report Measure rows in the overall column flag significant differences between time spent on individual SM platforms which are labeled ^A–D^ (Facebook ^A^, Twitter ^B^, Instagram ^C^, and Snapchat ^D^). The Mean^diff^ rows at the bottom of the table indicate significant mean differences between Objective and Self-Report Measures for each platform and overall (computed as Objective minus Self-report) are flagged. ** *p* < 0.01, *** *p* < 0.001 throughout).

**Table 2 ijerph-19-09847-t002:** Bivariate correlations between retrospectively self-reported and objectively assessed daily minutes spent on SM among all participants and self-reported users overall and by sex.

	Overall	By Sex		
	*R*	*R* Males	*R* Females	*Z* Sex *R*^diff^
Facebook	0.39 ***	0.43 ***	0.35 ***	0.81
Twitter	0.42 ***	0.51 ***	0.38 ***	1.39
Instagram	0.27 ***	0.44 ***	0.21 **	2.22 **
Snapchat	0.32 ***	0.57 ***	0.32 ***	2.70 **
All SM	0.26 ***	0.54 ***	0.16 *	3.79 ***

*Notes.* Z Sex *R*^diff^ refers to tests of the difference between the male and female specific correlation coefficients; * *p* < 0.05, ** *p* < 0.01, *** *p* < 0.001.

**Table 3 ijerph-19-09847-t003:** Multinomial regression models predicting the odds of self-report classification (relative to Accurate) for each SM platform as a function of participant sex and total number of SM platforms used.

Platform	Outcome	Predictors	OR	95% CI [OR]
Facebook				
	Under-estimator	Female Sex	0.89	0.39–2.03
		Number of SM platforms used	1.01	0.59–1.72
	Over-estimator	Female Sex	1.36	0.79–2.33
		Number of SM platforms used	0.96	0.69–1.35
Twitter				
	Under-estimator	Female Sex	1.08	0.44–2.67
		Number of SM platforms used	6.07 **	2.14–17.21
	Over-estimator	Sex	1.19	0.67–2.13
		Number of SM platforms used	7.94 **	4.10–15.37
Instagram				
	Under-estimator	Female Sex	1.22	0.64–2.31
		Number of SM platforms used	1.06	0.72–1.57
	Over-estimator	Sex	2.15 **	1.29–3.59
		Number of SM platforms used	1.51 **	1.08–2.10
Snapchat				
	Under-estimator	Female Sex	1.92 *	1.01–3.64
		Number of SM platforms used	1.69 *	1.08–2.66
	Over-estimator	Female Sex	2.26 **	1.35–3.79
		Number of SM platforms used	1.43 *	1.03–2.00
Total SM				
	Under-estimator	Sex	1.28	0.63–2.58
		Number of SM platforms used	1.38	0.91–2.08
	Over-estimator	Sex	1.43	0.78–2.62
		Number of SM platforms used	1.82 **	1.26–2.60

*Notes.* OR refers to Odds Ratio and 95% CI of [OR] refers the 95% confidence interval for the Odds Ratio; * *p* < 0.05, ** *p* < 0.01.

## Data Availability

Deidentified study data is available upon request from the corresponding author (email: sarah.boyle@lmu.edu).

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
