# Peer review of "Systematic Bias in Self-Reported Social Media Use in the Age of Platform Swinging: Implications for Studying Social Media Use in Relation to Adolescent Health Behavior"

_ijerph, 2022, doi:10.3390/ijerph19169847_

Round 1
Reviewer 1 Report
I enjoyed reading this article and found it easy to read and navigate.
The title includes platform swinging, should this phrase be mentioned in abstract and key words? It may be valuable to do this.
There is a line in discussion that needs to edited as here 'Concerningly, many participants in this study were not able to accurately'
The research discusses possible limitations- might the participants with the software use their devices differently as they aware their behaviours are being surveyed? Could this be discussed?
The authors highlight the relevance and significance of the work and the link back to the 2013 study highlights the developments offered by the current study.
Reviewer 2 Report
I thank the authors for conducting and reporting on this important study.
I found the study to be well-constructed at the conceptual, methodological and critical-reflective levels. Introduction: The authors have shifted an important knowledge gap and justified the study.
Methodology: It was clearly presented.
Conclusions: The authors explained the perspectives of the study, presenting the added value of the research
I would like the authors to clarify the following point:
, "For each platform and across platforms, participants who self-reported use within +/-9 minutes of their objective assessment are classified as "Accurate," those who self-reported time 10 or more minutes less than their objective assessment are classified as "Underestimators," and those who self-reported time exceeding their objectively logged time by 10 minutes or more are classified as "Overestimators. I am interested to know if the Authors referred to any specific classification? Were these their original findings?
I believe that the manuscript is suitable for publication with minor modifications.
Author Response
- I thank the authors for conducting and reporting on this important study.
We thank the reviewer for taking the time to review and provide us their feedback.
- I found the study to be well-constructed at the conceptual, methodological and critical-reflective levels.
Thank you.
- Introduction: The authors have shifted an important knowledge gap and justified the study.
Thank you.
- Methodology: It was clearly presented.
Thank you.
- Conclusions: The authors explained the perspectives of the study, presenting the added value of the research.
Thank you.
- I would like the authors to clarify the following point: "For each platform and across platforms, participants who self-reported use within +/-9 minutes of their objective assessment are classified as "Accurate," those who self-reported time 10 or more minutes less than their objective assessment are classified as "Underestimators," and those who self-reported time exceeding their objectively logged time by 10 minutes or more are classified as "Overestimators”. I am interested to know if the Authors referred to any specific classification? Were these their original findings?
No prior social media research had examined whether reporting inaccuracy is related to participant characteristics (sex, platform swinging) so there were no existing cut points to be used. As such, our choice to consider of 9 or fewer minutes off in either direction as “Accurate” and inaccuracy by 10 or more minutes in either direction “Overestimator/Underestimator” was informed by the degree of error that may be meaningful in the applied health research context as well as the distribution of reporting discrepancies. We have clarified this in the text on page 6 as pasted below:
“Cut-points selected to classify participants were informed by the distribution of reporting discrepancies in the sample as well as consideration of the degree of error that may be meaningful in the applied health research context.”
Note also that additional classification rationale was also provided on page 8 as pasted below:
“In contrast to raw discrepancies and correlations between measures, our classification of participants as accurate reporters, under-estimators, and over-estimators more meaningfully identified the proportions of participants who substantially and problematically mis-reported their SM time in either direction relative to those who were inconsequentially off only a handful of minutes.”
- I believe that the manuscript is suitable for publication with minor modifications.
Thank you. We share this perspective.